# A Comparative Study on Intention to Use Digital Therapeutics: MZ Generation and Baby Boomers’ Digital Therapeutics Use Intention in Korea

**DOI:** 10.3390/ijerph19159556

**Published:** 2022-08-03

**Authors:** Soojin Kim, Juhee Eom, Jiwon Shim

**Affiliations:** 1Division of Communication & Media, Ewha Womans University, Seoul 03760, Korea; sjinkim@ewha.ac.kr; 2Department of Law, Konkuk University, Chungju 05029, Korea; juheelight@gmail.com; 3Department of Philosophy, Dongguk University, Seoul 04620, Korea

**Keywords:** digital therapeutics, digital literacy, private concern, DTx use intention

## Abstract

Purpose: The aim of this study lies in articulating the relationship between digital literacy and private concern as a predictor of intention to use digital therapeutics. Materials and Methods: An online survey was conducted through a research company among 600 panels. The survey questionnaire consists of items of digital literacy, privacy concern, perceptions, and intention to use digital therapeutics, and the participants were asked to fill out the questions online. A structural equation model was established, and the difference in paths between the MZ generation and the baby boomers were examined. Results: Public perception of digital therapeutics was categorized into seven factors and the dimension of digital literacy as categorized into three factors. For the MZ generation, digital literacy and privacy concern both directly and indirectly affect the digital therapeutics use intention, in that higher the level of digital literacy and the lower the privacy concern, digital therapeutics perception and digital therapeutics use intention becomes intensified. For the baby boomers, digital literacy and privacy concern positively affect digital therapeutics perception, and as digital therapeutics perception becomes more positive, digital therapeutics use intention also increases. Direct effects of digital literacy and privacy concern to digital therapeutics use intention were not found for the baby boomers. Conclusion: In order to promote the use of digital therapeutics, it is common for the MZ generation and baby boomers to develop a positive perception toward digital therapeutics by cultivating digital literacy. For the MZ generation, privacy concerns need to be cautiously considered as they negatively affect the intention to use digital therapeutics.

## 1. Introduction

First introduced in 2000 by Seth Frank, digital health, largely encompassing media and internet-focused applications for the Internet, performs a few critical functions [1]. The exchanging of information, aiding informed decision making, promoting health, providing a means for information exchange and support, increasing self-care, managing demand for health services, and lowering direct medical costs are those critical functions of Internet-based application as a mean of digital health care.

Digital health care is a multidisciplinary umbrella term that includes concepts from an intersection between information technology and healthcare. Digital health care was born by the convergence of advanced information and communication technology and medical technology. As such, the core technologies of the fourth industrial revolution, such as artificial intelligence and big data, and the development of a digitally transformed society, are applied to the field of health care [2]. In digital health care, ‘digital’ refers to information and communication technology, and ‘health care’ refers to the overall health care industry, including pharmaceuticals, cosmetics, food, medical devices, medical information, health care systems, and general industry [3]. The broad scope of digital health includes categories such as mobile health (mHealth), health information technology (IT), wearable devices, telehealth and telemedicine, and personalized medicine [4].

Digital health care has a very high potential for development according to the growing interest in health care and the need for chronic disease management as an aging society deepens. Efforts to develop digital health care at the government level include the telemedicine service pilot project that started in 1988, the establishment of a national health computer network, and policies related to medical informatization such as dissemination of electronic medical records to medical institutions [5]. Recently, artificial intelligence has begun to be widely used in the medical field in public health such as detection and diagnosis of disease, administration of chronic disease, medical delivery systems, new drug development, and infectious disease detection [6].

Since artificial intelligence relies on digitized personal health data, the importance of data in digital health care is emerging in relation to the securing and utilization of these data [7]. In this context, the Korean government announced <Health Data F.LO.W. 2025> in June 2021. <Health Data F.LO.W. 2025> is related to the digital New Deal Policy announced by Korean government in 2020, and the Ministry of Health and Welfare will also prepare a basis for health and medical data utilization and develop digital health care using advanced technologies such as artificial intelligence and big data [8]. In accordance with the generation, collection, and use of this enormous amount of personal health data, the issue and interest in privacy protection increases.

Since individuals using digital health care have to utilize cutting-edge digital technology, digital literacy, which indicates how well they can handle digital technologies and acceptability of new digital technologies, is expected to affect the spread and utilization of digital health care. Digital healthcare has evolved, along with the development of digital technology. In this respect, it can be seen that individuals using digital health care presuppose the use of digital technology. Therefore, it can be assumed that digital literacy, which indicates how well you can handle digital technologies, will affect the dissemination and utilization of digital health care.

In this regard, it is necessary to pay attention to digital literacy related to the ability to handle digital health care. Digital literacy also affects health literacy, which accesses, understands, and utilizes health information, and can be measured by digital communication ability and creative production capacity using digital devices. Adoption of new digital technologies varies by age group [9]. In Korea, the generation called ‘Generation MZ’ refers to the generation that includes millennials (Generation M) born between 1980 and 1994; and Generation Z, born between 1995 and 2004 [10]. Baby boomers born between 1946 and 1964 are called digital immigrants because they were not born in the digital world but grew up to become accustomed to digital technologies. The MZ generation are called digital natives because they are familiar with the Internet and computers from birth and are free to use digital mobile devices. Because the baby boomers and MZ generations differ in their familiarity with digital devices and their literacy, it can be assumed that this will also affect the acceptability of digital healthcare among the MZ generation and baby boomers [11].

According to previous studies related to digital health acceptance, health interest, functional excellence and innovation, compatibility with other devices, design aesthetics, cost effectiveness, information quality, reliability of data security, and ease of use were related to the intention to accept digital health care services [12]. According to the research results, generally, trust in security has a significant effect on ease of use, as it will help users to quickly understand and use functions easily by allowing users to use products and services with confidence.

Since digital health care utilizes various data generated in an individual’s life, it is inevitable to avoid being connected to personal privacy. Depending on how much you care about privacy in a digital health care setting, it is necessary to review whether it affects access to and use of digital health care. However, there is still a lack of research on how concerns about privacy, such as security management of health care data, differ by generation and how this affects the acceptability of digital health care. Privacy issues such as data reliability and information security are considered to be important drivers for the development of digital health care, so research on them is very necessary. In addition, there are few studies showing how digital literacy, which plays an important role in understanding and applying digital technologies, is related to situations where people need to operate and utilize devices or services on their own, such as digital health care.

Therefore, in this study, we would like to examine how digital literacy and privacy concerns are related to the acceptance of digital technology. Specifically, we presented digital therapeutics as a typical digital health care, a technology that is expected to grow and attract attention in fields such as mental health, chronic disease, and pain control among digital health care. Digital therapeutics can be defined as a regulatory approved digital system or application that is prescribed to treat medical conditions, similar to that of new drug molecules or medical devices [13,14,15,16] Due to the nature of digital therapeutics, only patients who can operate and afford technological hardware may benefit leading to potential bias.

Studies have shown individuals of higher socioeconomic and education status tend to be healthier and have healthier behaviors [17,18,19] Age is also an important factor as older individuals may not own or be familiar using smart technologies (i.e., smartphones, tablets) [20]. Moreover, other identified barriers for older individuals include the complexity and lack of guidance when using these technologies [20]. The elderly may also not be interested in learning how to use new digital therapeutics interventions even if this could potentially be the population that would receive the largest benefit [21]. These older individuals would not need to physically visit their physician’s office for every appointment and could use remote monitoring as a tool to improve overall health.

Digital therapeutics are not yet popularized. Therefore, in this study, we first examine how the perception of digital therapeutics appears in Korea, and the effect of digital literacy and privacy concern on the perception and intention of using digital therapeutics.

## 2. Materials and Methods

### 2.1. Participants

Anonymous online surveys were conducted to collect data via a professional online research firm. The online research panel was recruited by random quota sampling, and the quota represented a general demographic composition similar to the most recent Korean Census Data [22]. The survey was administered online for a week from 1 July to 8 July 2021. A total of 600 participants were recruited. For the final data used, 306 (51%) were male and 294 (49%) were female. A total of 39 (6.5%) were teenagers, 101 (16.8%) were in their 20s, 98 (16.3%) were in their 30s, 128 (21.3%) were in their 40s, 130 (21.7%) were in their 50s, and 104 (17.3%) were in their 60s or older.

### 2.2. Measures

#### 2.2.1. Digital Literacy

The measuring items for digital literacy were adopted and modified from conceptualization of digital literacy in the work of Eshet-Alkalai and Eshet-Alkalai and Chajut [23,24]. A total of 16 items, including “I can send MP3 files to other people”, “I can upload my file on the internet web-hard”, “I can configure the web browser or explorer”, etc. were asked and used for factor analysis. Respondents were asked to answer the 16 items on a 5-point Likert scale ranging from 5 (strongly agree) to 1 (do not agree at all). Items measuring digital literacy and the results of the factor analysis are represented in Table 1.

#### 2.2.2. Privacy Concern

Privacy concern was measured by the items adopted and modified from the internet user’s information privacy concerns, which consist of three subdimensions of control, awareness of privacy practices, and collections [25]. Control dimension was measured using 3 items, including “consumer control of personal information lies at the heart of consumer privacy”, etc. (*α* = 0.77). Awareness of privacy practices dimension was measured using 3 items, including “a good consumer online privacy policy should have a clear and conspicuous disclosure”, etc. (*α* = 0.87). Collection dimension was measured using 4 items, including “it usually bothers me when online companies ask me for personal information” etc. (*α* = 0.79). The respondents were asked to answer on a 5-point Likert scale ranging from 5 (strongly agree) to 1 (do not agree at all). The coefficient *α* for 10 items of privacy concern was 0.89.

#### 2.2.3. Public Perception of Digital Therapeutics

A media content analysis was conducted as a pretest to develop items for public perception. We selected five Korean daily newspapers with high circulation and a range of readership profiles (http://www.bigkinds.or.kr). Of these, two are conservative, two are progressive, and the other is an industry-specialized newspaper. This method of newspaper selection has been used in other analyses of media reports to select a broad sample of newspapers with various readership profiles and political orientations [26,27].

Our search period was from 1 June 2020 to 1 June 2021, because this represented time of increasing academic and clinical interest in South Korea. Articles in the target newspapers were identified using the keyword “digital therapeutics”. This search identified more than a thousand articles and each article was scrutinized by one researcher from a team of three to assess whether the article met the inclusion criteria. The inclusion criteria were that digital therapeutics be the primary focus of article, and that simple publicity articles such as announcing or mentioning organizational personnel and single-photo articles with captions only were excluded. As a result, 680 articles were identified as eligible for the analysis.

The next step was to construct survey items from the 680 articles. Each article was again scrutinized by two researchers and 38 items representing public perceptions were extracted. The respondents were asked to answer the 38 items using five-point scales anchored with “strongly disagree” and “strongly agree”. Items measuring public perceptions of digital therapeutics were analyzed by the explanatory factor analysis and are represented in Table 2.

#### 2.2.4. Digital Therapeutics Use Intention

The measuring items for digital therapeutics use intention were adopted from the work of Venkatesh and Davis [28]. The items include “I have an intention to use digital therapeutics”, “I would speak positively of digital therapeutics”, “I would recommend to others to use digital therapeutics”, and “I intend to use digital therapeutics continuously”, on a five-point scale anchored with “strongly disagree” and “strongly agree.

### 2.3. Ethical Considerations

This study was approved by Institutional Review Board of Dongguk University (DUIRB-202109-21).

### 2.4. Statistical Analysis

Statistical analyses were performed in three stages. In the first stage, exploratory factor analyses were employed to identify the underlying relationships between measured variables (digital literacy and digital therapeutics perception) to extract their common dimensions. In the second stage, a reliability test was performed to assess the stability and consistency of the measured items, including descriptive statistics. In the third stage, a structural equation model was established to examine the relationship between latent variables, and moderation effect analysis was conducted to examine the difference between the millennials and baby boomers.

## 3. Results

### 3.1. Reliability and Validity of the Variables

The means, standard deviation, and Pearson correlations of the observed variables are presented in Table 1. For the reliability analysis, Cronbach’s *α* test was conducted several times. First, Cronbach’s *α* test was conducted for all items of the constructs. The variables from the latent constructs used summated rating scales composed of many measurement items extracted from the exploratory factor analysis. All the items consisting of observable variables reported Cronbach’s *α* coefficients of over 0.60.

### 3.2. The Types of Digital Literacy

As a result of an exploratory factor analysis for the items extracted from Eshet-Alkalai and Eshet-Alkalai and Chajut’s studies [23,24], 16 items of measuring digital literacy were categorized into three factors. The EFA showed 16 factors, explaining 68.79% of total variance (Table 2).

The first factor, named as information utilization ability, was found to explain 30.79% of the total variance. This factor represented one’s ability to use the Internet. The second factor, named as creative producing ability, was found to explain 19.87% of the total variance. This factor was related to one’s creative ability to produce and manage one’s Internet space and to show oneself on the Internet. The third factor, named as digital communication ability, was found to explain 18.12% of the total variance. This factor represented one’s communication ability to send messages online by email and use the Internet to know other people. Specific items consisting of each factor are shown in Table 2.

### 3.3. Public Perceptions of Digital Therapeutics

As a result of an exploratory factor analysis for the items extracted from media content analysis, public perceptions of digital therapeutics were categorized into seven factors. The EFA showed 32 factors, explaining 60.54% of total variance (Table 3).

The first factor, named as regulation need, was found to explain 16.35% of the total variance. This factor represented the public’s legal requirements related to the use of digital therapeutics. The second factor, named as cost efficiency, was found to explain 10.72% of the total variance. This factor was related to public tendency to judge the efficiency of the digital therapeutics. The third factor, named as health benefit, was found to explain 9.60% of the total variance. This factor represented whether the use of digital therapeutics can benefit one’s own health. The fourth factor, named as medical concern, was found to explain 7.53% of the total variance. This factor represented public thought on how medical staff related to digital therapeutics think of digital therapeutics use in medical encounters. The fifth factor, named as device credibility, was found to explain 6.00% of the total variance. This factor is related to one’s favorability toward direct communication with doctors and represents trustworthiness of doctor’s medical device. 

The sixth factor, named as health inequality, was found to explain 5.73% of the total variance. This factor represented one’s concern that using digital therapeutics would result in any inequality related to health. The last factor was named uncertain literacy, in that it represented one’s idea that understanding and utilizing digital therapeutics is not an easy matter.

### 3.4. The Difference between the MZ Generation and the Baby Boomers

After the reliability and validity test, the group analysis between the MZ generation and the baby boomers was completed. The MZ generation showed a higher mean value in health literacy, while baby boomers showed a relatively higher mean value in privacy concern. For digital therapeutics perception, the MZ generation showed a higher mean value in cost efficiency, device credibility, health inequality, and uncertain literacy, while baby boomers showed a higher mean value in regulation need and health benefit. For medical concern, the MZ generation and baby boomers showed the same mean value, and there were almost no differences in the mean value of digital therapeutics use intention in two groups (Table 4).

#### 3.4.1. The Relationship between Digital Literacy, Privacy Concern, DTx Perception, and DTx Use Intention

To find out the relationship between digital literacy, privacy concern, digital therapeutics perception, and digital therapeutics use intention, we proposed a structural equation model. We evaluated the goodness-of-fit indices for the proposed structural equation model and the model generated χ^2^ = 582.885, df = 107, *p* = 0.000; and the other fit index indicated a good fit within the accepted exhortation level (CFI = 0.904, IFI = 0.905, RMSEA = 0.086).

The result of path analysis showed that digital literacy positively affected digital therapeutics perception (*ß* = 0.250, *p* < 0.001), in that as digital literacy increases, digital therapeutics perception intensifies. However, the direct effect of digital literacy to digital therapeutics use intention was not found.

Moreover, the direct effect of privacy concern to digital therapeutics perception was found to be statistically significant (*ß* = 0.450, *p* < 0.001), in that as privacy concern increases, digital therapeutics perception also intensifies. In the case of digital therapeutics use intention, the negative effect of privacy concern was found to be statistically significant (*ß* = −0.138, *p* < 0.001). Thus, as privacy concern increases, the intention of digital therapeutics use decreases.

#### 3.4.2. The Path Difference between the MZ Generation and the Baby Boomers

In the main research model, digital literacy did not affect DTx use intention. To compare the MZ generation and baby boomers, a moderation model analysis was performed. We again evaluated the goodness-of-fit indices for the proposed structural equation model for the moderation analysis, and the moderation model generated χ^2^ = 575.991, df = 214, *p* = 0.000; and the other fit index indicated a good fit within the accepted exhortation level (CFI = 0.900, IFI = 0.901, RMSEA = 0.064).

In the moderation model where the comparative analysis of the MZ generation and baby boomers was carried out, all the paths turned out to be statistically significant in the MZ generation.

First of all, the direct effect of digital literacy toward digital therapeutics use intention was found to be statistically significant in the MZ generation (*ß* = 0.156, *p* < 0.01). In the MZ generation, digital literacy not only directly affected digital therapeutics use intention, but also mediated digital therapeutics perception (*ß* = 0.305, *p* < 0.01) and affected digital therapeutics use intention significantly (*ß* = 0.879, *p* < 0.01).

Secondly, privacy concern positively affected digital therapeutics perception (*ß* = 0.444, *p* < 0.001) and negatively affected DTX use intention (*ß* = −0.203, *p* < 0.001) (Figure 1).

In baby boomers, direct effects of digital literacy and privacy concern to digital therapeutics use intention were not found to be statistically significant. A mediated effect of digital therapeutics perception was found, in that digital literacy mediated digital therapeutics perception (*ß* = 0.267, *p* < 0.05) and affected digital therapeutics use intention (*ß* = 0.877, *p* < 0.001).

In baby boomers, a mediation effect also appeared, in that as privacy concern increased, digital therapeutics perception intensified (*ß* = −0.138, *p* < 0.001), and digital therapeutics perception affected digital therapeutics use intention (Figure 2).

## 4. Discussion

This study was conducted to investigate how digital literacy and privacy concerns affect digital healthcare acceptance. In particular, we tried to predict the acceptability of digital healthcare by examining the characteristics of digital literacy and people’s perceptions of digital therapeutics, which are a part of digital healthcare.

The main findings of this study are summarized as follows. First, it is necessary to positively change the perception of digital health care in order to increase the acceptance of digital health care. According to the results of this study, perceptions of digital therapeutics were divided into seven categories, including legal dimensions, cost effectiveness dimensions, health dimensions, device dimensions, and so on. Therefore, taking these individual dimensions into account, it is not easy to convert and reinforce all perceptions positively. Nevertheless, efforts such as adding technical supplementation to the digital treatment machine to increase credibility in the device is sufficiently possible. The device credibility is the part where the perception gap of MZ generation and the baby boomers existed, so it can be said that it is a part that needs to be strategically paid special attention to in order to increase the acceptance of digital healthcare.

A second interesting finding is related to privacy concerns. In the MZ generation, it was found that the greater the concern about personal information, the lower the DTx use intention. On the other hand, in baby boomers, privacy concerns did not appear to affect the DTx use intension. These results do not appear because the MZ generation is more concerned about privacy than the baby boomers in digital healthcare. Rather, privacy concerns are higher among baby boomers than the MZ generation. However, it can be inferred from the result that baby boomers are relatively vulnerable to disease situations because they are older than the MZ generation, and therefore have a high degree of involvement in digital health care.

In the case of the MZ generation, there are various ways to increase the intention to use DTx. By cultivating digital literacy and devising ways to overcome privacy concerns, the perceptions of digital therapeutics can be positively reinforced, thereby increasing the intention to use DTx. However, as privacy concerns increase, the intention to use DTx decreases, so this is an area that needs to be carefully reviewed.

In the case of baby boomers, the pathways to increase the intention to use digital therapeutics are not as diverse as the MZ generation. The intention to use DTx can be increased only by positively reinforcing the awareness of digital therapeutics through measures to cultivate digital literacy and overcome privacy concerns. Therefore, it is important to cultivate the literacy necessary for the digital environment and to understand privacy-related matters through education.

The limitations of this study are as follows. First, though the explanation of DTx was provided in the survey, the interpretation and understanding of DTx may differ for each respondent. Second, the items measuring DTx perception in this study were extracted from media analysis. In the future, it is necessary to increase the acceptance of digital healthcare by specifying and diversifying the perception of digital therapeutics through an expert group interview.

Although this study reviewed the relationship between perception of digital therapeutics, digital literacy, and privacy concerns in the intention to use digital therapeutics, future research should consider the relationship of various variables that can influence this process. For example, illness experience or the need for face-to-face treatment may affect perceptions of digital therapeutics. Cultural variables may also be a factor influencing perceptions and intention to use digital therapeutics.

## 5. Conclusions

Depending on how the perception of digital therapeutics is formed, the intention to use digital therapeutics may vary. According to the results of this study, perceptions of digital therapeutics were divided into seven categories, including legal requirements related to digital therapeutics, cost efficiency, health benefit, medical concern, device credibility, health inequality, and uncertain literacy. These dimensions contain both positive expectations and concerns, and ultimately affects the intention to use digital therapeutics.

In addition, factors affecting the perception of digital therapeutics across generations are digital literacy and privacy concerns. In this point, it is necessary to cultivate digital literacy for a better understanding of digital therapeutics. At the same time, it is necessary to increase accessibility to digital therapeutics by strengthening the legal and institutional environment for privacy concerns, such as personal information protection.

In fact, discussing how digital therapeutics might affect human lives is beyond the scope of this study. However, understanding people’s perceptions of digital therapeutics is a necessary part of understanding our lives where many of which are already digitalized. Ultimately, understanding people’s perceptions of digital therapeutics can be a starting point for disease treatment and health promotion.

## Figures and Tables

**Figure 1 ijerph-19-09556-f001:**
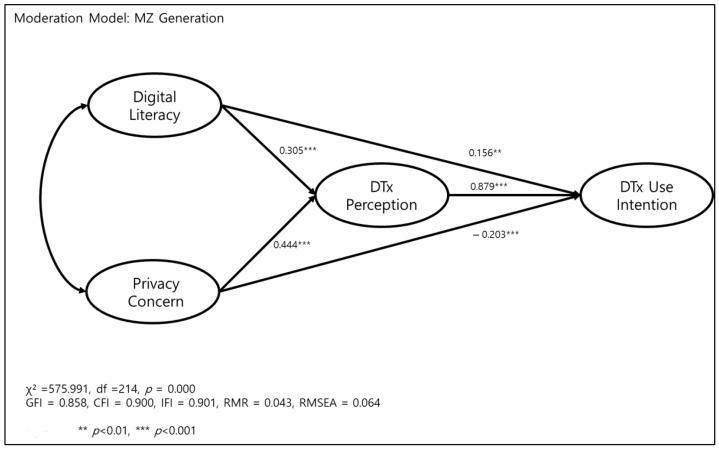
Effect of digital literacy and privacy concern on digital therapeutics (DTx) perception and digital therapeutics (DTx) use intention in MZ generation.

**Figure 2 ijerph-19-09556-f002:**
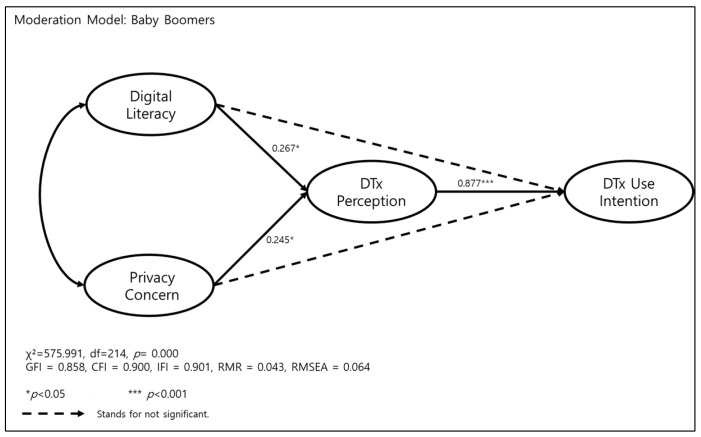
Effect of digital literacy and privacy concern on digital therapeutics perception and digital therapeutics (DTx) use intention in baby boomers.

**Table 1 ijerph-19-09556-t001:** Means, standard deviations, and Pearson correlations among predictors of DTx use intention.

	M	SD	(1)	(2)	(3)	(4)
(1) Digital literacy	3.45	0.60	1			
(2) Privacy concern	3.98	0.58	0.174 **	1		
(3) DTx perception	3.53	0.34	0.273 **	0.502 **	1	
(4) DTx use intention	3.51	0.68	0.317 **	0.195 **	0.435 **	1

(N = 600) ** *p* < 0.01

**Table 2 ijerph-19-09556-t002:** The result of exploratory factor analysis: digital literacy.

	Factors		
	Information Utilization Ability	Creative Producing Ability	Digital Communication Ability
I can send MP3 files to others.	0.874	0.126	0.127
I can upload my files to web-hards on the internet, etc.	0.851	0.218	0.054
I can configure the web browser (Explorer, etc.).	0.848	0.161	0.133
I use the Internet well for my work/class.	0.844	0.168	0.161
I can find the information I need through a search engine.	0.764	−0.055	0.249
I can understand exactly what the information conveys.	0.759	0.124	0.233
I can express what I want to say through pictures or videos on the Internet.	0.640	0.409	0.297
As a fan of celebrities such as stars, sportsmen, politicians, and CEOs, I can produce fan art such as videos, audio, and pictures.	0.026	0.877	0.078
I can parody or remix drama music videos, popular songs, and movies with audio and video.	0.072	0.866	0.076
I decorate and manage my blog or mini-homepage well.	0.168	0.718	0.252
I am good with digital cameras and camcorders.	0.436	0.640	0.127
I express my intentions clearly in writing on the Internet.	0.255	0.556	0.410
I use the Internet to send group messages or texts to my friends.	0.117	0.213	0.795
I use the internet to send my regards to my friends.	0.170	0.242	0.795
By using the Internet, I have come to understand myself and others better.	0.124	0.140	0.781
I think the Internet provides an important opportunity to get to know a variety of people from different regions.	0.301	0.003	0.707
Kaiser–Meyer–Olkin (KMO): 0.894, Bartlett’s Test of Sphericity: 6421.587, df = 120, *p* = 0.000			
Eigen Value	4.927	3.181	2.899
% of Variance Explained	30.795	19.879	18.122
Cumulative Variance Explained	30.795	50.673	68.795
Cronbach’s α	0.927	0.845	0.825
M	3.954	2.832	3.589

**Table 3 ijerph-19-09556-t003:** The result of exploratory factor analysis: digital therapeutics (DTx).

	Factors						
	Regulation Need	Cost Efficiency	Health Benefit	Medical Concern	Device credibility	Health Inequality	Uncertain Literacy
To use digital therapeutics, guidelines for cybersecurity permission and examination of medical devices must be prepared.	0.835	0.113	0.208	−0.045	0.078	0.052	−0.092
Personal information protection regulations for the use of digital therapeutics should be strengthened.	0.805	0.118	0.137	0.025	0.111	0.038	−0.034
To use digital therapeutics, screening criteria such as permission for medical devices applied with big data, VR, and AI technologies must be clear.	0.797	0.110	0.246	−0.050	0.055	0.014	−0.064
For digital therapies, the stability of technology is the most important.	0.712	0.227	−0.041	0.156	−0.156	0.084	0.111
Expectations for digital therapeutics are high, but there is still a long way to go in terms of licensing, insurance, and data utilization.	0.686	0.033	0.025	0.089	0.046	0.140	0.276
The interest and support of the government are essential for the development of digital therapeutics.	0.589	0.380	0.123	0.129	−0.091	0.081	0.129
As we enter an aging society, disease management programs such as digital treatments will be more necessary.	0.540	0.395	0.285	−0.024	−0.116	0.175	−0.141
For digital therapeutics, the effectiveness of technology (therapeutic effect) is the most important.	0.513	0.274	0.122	0.183	−0.141	0.059	0.206
If it is possible to claim private health insurance, I am willing to use digital therapeutics.	0.280	0.747	0.154	0.077	−0.087	−0.036	0.096
If it is digital therapeutics developed by conglomerates, I am willing to use it.	0.030	0.714	0.258	0.078	−0.004	0.067	0.142
I am willing to use digital therapeutics if they are not expensive.	0.266	0.710	0.225	0.010	−0.150	0.017	0.018
If digital treatment is successful in treating obesity, I am willing to use digital therapeutics.	0.227	0.677	0.089	0.098	0.059	0.205	−0.101
If medical efficacy has been proven, I am quite positive about digital therapeutics.	0.343	0.536	0.371	−0.008	−0.121	0.051	−0.131
People who know how to use digital therapeutics will be healthier.	−0.014	0.516	0.196	−0.027	0.019	0.504	−0.085
Digital therapeutics can treat medical diseases and disability.	0.003	0.188	0.818	0.065	−0.127	0.079	0.039
Digital therapeutics can treat medical disorders.	0.057	0.139	0.797	0.081	−0.091	0.029	0.038
You can see your health condition through digital therapeutics.	0.260	0.299	0.639	−0.003	−0.033	0.066	0.009
Digital therapeutics help manage and prevent chronic diseases.	0.431	0.186	0.579	−0.051	−0.061	0.026	−0.131
Digital therapies can get help optimizing medication (taking your medication) through digital treatment.	0.376	0.273	0.578	−0.023	−0.008	0.025	−0.132
If doctors are reluctant to prescribe digital therapeutics, it may be because they are concerned about the side effects of using digital devices themselves.	0.155	0.029	0.025	0.742	0.103	0.027	0.033
If doctors are reluctant to prescribe digital therapeutics, the reason may be that the treatment effect has not been sufficiently proven.	0.223	0.053	−0.078	0.709	0.049	−0.184	0.110
If doctors are reluctant to prescribe digital therapeutics, the reason may be that they do not expect patients to use digital devices well.	−0.031	0.045	0.086	0.643	0.115	0.237	0.097
If doctors are reluctant to prescribe digital therapeutics, it is probably because they are not used to using digital devices.	−0.062	0.120	−0.016	0.510	0.112	0.397	0.020
If doctors are reluctant to prescribe digital therapeutics, it may be because they are not yet covered by health insurance or indemnity insurance.	0.071	0.131	0.035	0.447	0.034	0.404	−0.278
If doctors are reluctant to prescribe digital therapeutics, it may be because they think it is inhumane or unethical to use digital devices for treatment.	−0.231	−0.101	0.195	0.445	0.351	0.158	0.280
Digital therapeutics are unlikely to be as effective as those using conventional drugs and medical devices.	0.077	0.045	−0.136	0.063	0.796	−0.071	−0.020
I do not trust the use of digital therapeutics without seeing a doctor or using digital therapeutics without a doctor’s prescription.	0.126	−0.124	−0.195	0.148	0.713	0.029	0.107
I think it is inhumane to treat people through digital devices without seeing a doctor.	−0.261	−0.145	0.041	0.253	0.626	0.031	0.212
If the use of digital therapeutics become active, there will be a health gap between those who can use them and those who don’t.	0.138	−0.007	0.091	0.154	0.007	0.736	0.194
If the use of digital therapeutics become active, there will be differences in patient preferences between hospitals that prescribe digital therapeutics and those that do not.	0.343	0.191	0.000	0.095	−0.105	0.651	0.166
It is not an easy task to learn and use the use of digital therapeutics.	0.090	0.046	−0.042	0.114	0.204	0.172	0.762
There are many problems to be solved before getting used to digital therapeutics.	0.535	0.069	−0.083	0.162	0.073	0.076	0.553
Kaiser–Meyer–Olkin (KMO): 0.870, Bartlett’s Test of Sphericity: 3685.262, df = 55, *p* = 0.000							
Eigen Value	5.234	3.431	3.072	2.410	1.922	1.834	1.470
% of Variance Explained	16.356	10.721	9.600	7.532	6.007	5.731	4.595
Cumulative Variance Explained	16.356	27.078	36.677	44.209	50.216	55.947	60.542
Cronbach’s α	0.890	0.826	0.826	0.699	0.645	0.664	0.642
Mean	4.07	3.64	3.49	3.33	3.00	3.58	3.60

**Table 4 ijerph-19-09556-t004:** The means and standard deviations of variables: MZ generation and baby boomers.

Variables	Dimensions	Groups	M	SD
Health Literacy	Information utilization ability	MZ	4.14	0.71
		Baby boomers	3.81	0.69
	Creative production ability	MZ	3.04	0.86
		Baby boomers	2.67	0.76
	Digital communication ability	MZ	3.62	0.76
		Baby boomers	3.56	0.68
Privacy Concern	Collection	MZ	3.85	0.69
		Baby boomers	3.92	0.62
	Control	MZ	3.93	0.66
		Baby boomers	3.93	0.61
	Awareness	MZ	4.09	0.74
		Baby boomers	4.15	0.66
DTx Perception	Regulation need (dtxF1)	MZ	4.04	0.62
		Baby boomers	4.09	0.54
	Cost efficiency (dtxF2)	MZ	3.66	0.60
		Baby boomers	3.62	0.55
	Health benefit (dtxF3)	MZ	3.46	0.58
		Baby boomers	3.50	0.55
	Medical concern (dtxF4)	MZ	3.33	0.60
		Baby boomers	3.33	0.54
	Device credibility (dtxF5)	MZ	30.08	0.70
		Baby boomers	2.94	0.63
	Health inequality (dtxF6)	MZ	3.62	0.70
		Baby boomers	3.55	0.65
	Uncertain literacy (dtxF7)	MZ	3.64	0.66
		Baby boomers	3.57	0.63
DTx Use Intention		MZ	3.50	0.70
		Baby boomers	3.52	0.67

## Data Availability

All data will be available upon requested.

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
