# Peer review of "A Comparative Study on Intention to Use Digital Therapeutics: MZ Generation and Baby Boomers’ Digital Therapeutics Use Intention in Korea"

_ijerph, 2022, doi:10.3390/ijerph19159556_

Round 1

Reviewer 1 Report

   The submitted article conducts an online survey on which a thorough statistical analysis is performed about the difference in Digital Therapeutics usage between two generations.

      There are several aspects that should be addressed before consideration for publication.

      On page 4, lines 163 and following. More information on the factors listed in Table 3 should be shown. The 38 items evaluated should be analyzed in more detail. Some of them could be highly correlated (questions 3 and 7 on page 7?). Some questions are open to interpretations that would make it difficult to extract information from the answers. This would need to be dealt with in a more rigorous way.

      On page 14, line 315 and following. The authors seem to side with improving the perception of digital health care. This should be treated with more care as it seems to assume a correct answer in what is an open and nuanced debate.

      It appears that the authors assume that the two factors studied are the most significant. Perhaps other factors should be taken into account. For example, having suffered from a serious illness or the need for face-to-face treatment may affect these perceptions. Cultural level may also be a factor influencing intentions and even applicability of this technology.

Reviewer 2 Report

The study aims at understanding the relationship between digital literacy and private concern as predictor of digital therapeutics use. The study is based on an online questionnaire and results seem promising. The overall paper is well structured, methodology well explained as well as the data analysis performed. Moreover, English is proficient. The subject of the study is quite interesting and I think of interest for the research communities of Digital Health.

I would suggest to authors to clearly highlight the limits of the study and further improvements needed.

Author Response

Thank you for your interest in this study and encouraging us by giving a kind review. Based on the reviewer's suggestion, research limitations and implications for future study have been added. Once again, we sincerely thank you for giving us a good insight into our research. Additional limitations of the study and revised parts are written in blue colors in the manuscripts. Many thanks again for your insightful suggestions.

This manuscript is a resubmission of an earlier submission. The following is a list of the peer review reports and author responses from that submission.